# Eradication of *Saccharomyces*
*cerevisiae* by Pulsed Electric Field Treatments

**DOI:** 10.3390/microorganisms8111684

**Published:** 2020-10-29

**Authors:** Efrat Emanuel, Irina Dubrovin, Ester Hanya, Gad A. Pinhasi, Roman Pogreb, Rivka Cahan

**Affiliations:** 1Department of Chemical Engineering and Biotechnology, Ariel University, Ariel 40700, Israel; efiem80@gmail.com (E.E.); Irinadu@ariel.ac.il (I.D.); esterhanya@gmail.com (E.H.); gadip@ariel.ac.il (G.A.P.); 2Department of Physics, Ariel University, Ariel 40700, Israel; ptoman@ariel.ac.il

**Keywords:** pulsed electric fields, current density, conductivity, eradication, yeast

## Abstract

One of the promising technologies that can inactivate microorganisms without heat is pulsed electric field (PEF) treatment. The aim of this study was to examine the influence of PEF treatment (2.9 kV cm^−1^, 100 Hz, 5000 pulses in trains mode of 500 pulses with a pulse duration of 10 µs) on *Saccharomyces cerevisiae* eradication and resealing in different conditions, such as current density (which is influenced by the medium conductivity), the sort of medium (phosphate buffered saline (PBS) vs. yeast malt broth (YMB) and a combined treatment of PEF with the addition of preservatives. When the *S. cerevisiae* were suspended in PBS, increasing the current density from 0.02 to 3.3 A cm^−2^ (corresponding to a total specific energy of 22.04 to 614.59 kJ kg^−1^) led to an increase of *S. cerevisiae* eradication. At 3.3 A cm^−2^, a total *S. cerevisiae* eradication was observed. However, when the *S. cerevisiae* in PBS was treated with the highest current density of 3.3 A cm^−2^, followed by dilution in a rich YMB medium, a phenomenon of cell membrane resealing was observed by flow cytometry (FCM) and CFU analysis. The viability of *S. cerevisiae* was also examined when the culture was exposed to repeating PEF treatments (up to four cycles) with and without the addition of preservatives. This experiment was performed when the *S. cerevisiae* were suspended in YMB containing tartaric acid (pH 3.4) and ethanol to a final concentration of 10% (*v/v*), which mimics wine. It was shown that one PEF treatment cycle led to a reduction of 1.35 log10, compared to 2.24 log10 when four cycles were applied. However, no synergic effect was observed when the preservatives, free SO_2_, and sorbic acid were added. This study shows the important and necessary knowledge about yeast eradication and membrane recovery processes after PEF treatment, in particular for application in the liquid food industry.

## 1. Introduction

Microorganism sterilization is an essential process in a variety of industries and medical applications. The available methods include thermal treatments, filtration, preservative addition, ultraviolet (UV) radiation, pulsed light (PL), cold plasma (CP), and pulsed electric fields (PEFs). Thermal treatments are the common method for inactivating microorganisms in the food industry, and there are comprehensive data on the available thermal processes [1]. However, a variety of food products are damaged, and their quality is decreased by exposure to thermal treatments. Wine is an example of a food product in which quality is impaired when sterilized by thermal treatments. 

PEF is a promising technology that can inactivate microorganisms without heat [2,3,4,5]. This method is regarded as a “clean” technology that does not change food flavor, taste, or color [6]. Microorganism eradication using PEF was previously reported in processing nectar [7], milk [8], liquid eggs [9], and wine [10]. PEF technology applies high electric-field pulses of short duration, generally between 1–100 µs [11,12]. The range of electric field is from 1 to 100 kV cm^−1^ [11,12,13,14,15]. The pulse shape can be a square or exponential pattern [7,16], and the number of pulses can range from one to tens of thousands [12,17,18]. The extent of microorganism eradication depends on the applied electric parameters, the treatment chamber materials and configuration, the cell size and type, the medium osmolarity, pH, and electrical conductivity [19,20,21,22,23].

It has been shown that electroporation causes pore formation and rapid breakdown of cell membrane, leading to membrane permeability [12,19]. Based on theoretical and experimental studies, this phenomenon occurs when the electric field exceeds the natural cell membrane potential. In relation to the treatment parameters mentioned above, electroporation can be divided into four ranges by its properties: irreversible with thermal damage, nonthermal irreversible, reversible, and no detectable electroporation. (1) Irreversible electroporation that is accompanied by thermal injury is characterized by protein denaturation and an imbalance of molecules released [24]. (2) Nonthermal irreversible electroporation is characterized by pore formation and slow cell resealing; or not resealing at all, leading to a release of cell contents [25]. (3) Reversible electroporation is characterized by pore and molecule movements in and out of the cell. However, a pore resealing can occur under appropriate environmental conditions, where the majority of the electroporated cells retain their viability. Pore formation may occur in less than a second, while resealing takes place in a range of minutes [26,27,28]. (4) Below a certain PEF strength, there is no detectable electroporation, regardless of the applied electric field and its duration [29,30].

PEF-treated microorganism viability is generally estimated based on the ability to multiply. The absence of multiplication is regarded as nonexistence of microbial life [31]. However, multiplication depends on various stress and environmental conditions [32]. *Live* or *dead* microorganism designation is not clear, since the process from cell life to death, and the reverse route of recovery, are not fully understood [1,33]. Sublethally injured microorganisms, as well as viable but nonculturable (VBNC) cells, are important forms of life that may be induced by environmental and stress conditions such as hydrostatic pressure, heat treatment, pulsed light, cold plasma, ultraviolet radiation, and PEF treatment [1]. The recovery of sublethally injured yeast and VBNC cells may take place under suitable conditions [34].

In applications of microbiological stability, the addition of preservatives is an essential step following the disinfection treatments. Preservatives can prevent the recovery of the sublethally injured cells and the proliferation of VBNC microorganisms. 

In food industries, sulfur dioxide (SO_2_) is a common preservative and is used in particular for winemaking [4,35]. The advantages of SO_2_ as a preservative include its ability to serve as an active antioxidant that inactivates certain microorganisms. In addition, it has a positive effect on some organoleptic characteristics such as aroma complexity and color stability [35]. The disadvantages of SO_2_ are that it may neutralize aromatic compounds [36] and affect human health by causing allergic reactions or sensitivity [37]. Thus, many efforts are being made to find alternatives in order to minimize the use of SO_2_. Another common preservative is sorbic acid, and some of its salts (in particular potassium sorbate) are already widespread in food processing and pharmaceuticals. In general, sorbates have been widely accepted as effective yeast and mold inhibitors. In contrast to SO_2_, sorbic acid is known as a preservative with a good biocompatibility and safety profile. Varied products in the pharmaceutical and food industries with a high percentage of water, such as suspensions, emulsions, aqueous solutions, or gels, can be preserved by means of the physical or chemical interactions of sorbic acid and its salts [38,39]. 

In the liquid food industry, control of yeast spoilage is necessary [40]. *Saccharomyces cerevisiae* is known as an important microorganism for beverage production [41,42]. Compared to other microorganisms, *S*. *cerevisiae* proliferate at low pH levels and high alcohol concentrations. However, at the end of a beverage production process such as wine, thermal treatment is applied to avoid yeast multiplication [43]. 

In addition, there are wines, for example, classic sparkling-wine, for which, in the final production step, the yeast biomass has to be removed. Separation is mostly based on rotating and inclining the bottle until all the yeast cells settle into the bottle’s neck. This method takes about 60 days [36,44]. Pretorius, 2000, described immobilization of the yeast (10^9^ cells g^−1^) in beads composed of 2% calcium alginate. This method shortened the time of yeast separation and reduced energy consumption [45]. Berovic et al. (2014) developed a rapid magnetic yeast separation in sparkling wine. The cells were absorbed to magnetized nanoparticles of iron oxide maghemite, including amino groups with a positive charge that promote electrostatic absorption of the yeast cells’ negatively charged surfaces. This method led to complete yeast separation within 15 min [44]. PEF can serve as an alternate technology that avoids the negative effects of the heat treatment.

The aim of this work was to study the effect of electric-current density (as a consequence of the medium’s conductivity) on yeast viability and membrane permeability. The recovery of the cells was examined in different media, phosphate buffered saline (PBS) or rich yeast malt broth (YMB). In addition, the viability of the yeast was examined after applying a combination of PEF technology and the addition of preservatives. This investigation was conducted on media that mimics wine characteristics, with different preservative concentrations. Our study shows the important and necessary knowledge about the yeast membrane recovery process after PEF treatment, in particular for application in the liquid food industry. 

## 2. Materials and Methods

### 2.1. Growth Conditions

*S. cerevisiae* var. *bayanus* (*S. cerevisiae*) yeast (Lallemand Inc., Montréal, QC, Canada) was suspended in water, spread on a yeast malt (YM) (Becton Diskinson, Franklin Lakes, NJ, USA) agar plate, and incubated for 24 h at 37 °C. An isolated colony was taken and suspended in 12.5 mL yeast malt broth (YMB) with agitation at 150 rpm at 37 °C, until it reached the mid-log phase. The culture was then centrifuged at 4000 g for 10 min at 4 °C (Avanti J-E centrifuge, Beckman Coulter, Carlsbad, CA, USA), and the sediment was washed with ultrapure (UP) water having a resistance of 18.4 MΩ-cm at 25 °C (Synergy UV water purification system, Merck, Darmstadt, Germany). The yeast sediment was suspended in UP water and in different concentrations of PBS to a final optical density of 0.01 at 600 nm, determined by spectrophotometer (Genesys 10S UV-VIS, Thermo Scientific, Waltham, MA, USA).

For the second part of this research, the *S. cerevisiae* yeast was grown in YMB with tartaric acid (pH 3.4) until it reached the mid-log phase. Increasing ethanol concentrations were then added to the culture over a period of 5 h to achieve a final concentration of 10% ethanol (*v/v*). After 2 h, the multiplication of *S. cerevisiae* yeast was examined by spectrophotometer. 

The YMB medium with the tartaric acid and the ethanol (10%) at pH 3.5 was used to mimic wine; it was therefore designated as YMBW (yeast malt broth wine). The *S. cerevisiae* were diluted in the YMBW medium to a final concentration of 0.02 OD at 660 nm.

#### 2.1.1. PBS Solution

10 mM PBS (REF: 02-023-1A) was purchased from Biological Industries, Kibbutz Beit-Haemek, Israel. The PBS components were: 1.47 mM KH_2_PO_4_, 8.1 mM Na_2_HPO_4_, 2.67 mM KCl, and 136.9 mM NaCl. Different PBS concentrations (0.089–0.54 mM) were obtained by dilution with UP water.

#### 2.1.2. Preservative Solutions

Sorbic acid and free SO_2_ were prepared by dissolving potassium sorbet and potassium metabisulfite, respectively. The solubility of these materials depends on the medium properties: pH, temperature, ethanol, and sucrose concentration. Hence, the required mass of sorbic acid was multiplied by a factor of 1.35, and the required mass of free SO_2_ was multiplied by a factor of 1.72. For preparing high-concentration solutions, 118 g of potassium sorbet, 163 g of potassium metabisulfite, and 0.3 g of tartaric acid were dissolved in UP H_2_O (100 mL final volume); the final concentrations of free SO_2_ and sorbic acid were 875 ppm each, with a pH of 3.4. The same procedure was followed to achieve a final concentration of 1750 ppm free SO_2_ and 4381 ppm sorbic acid. Only 5 µL of these solutions were inserted into 170 µL treated culture. Thus, the final concentrations during the experiments were 25 ppm free SO_2_ and 25 ppm sorbic acid, or 50 ppm free SO_2_ and 125 ppm sorbic acid.

#### 2.1.3. The Solutions’ Conductivity and pH

The conductivity of PBS (155–1050 µS cm^−1^), UP water (1 µS cm^−1^), and YMBW, and (1200 µS cm^−1^). Conductivity was measured before the yeast inoculation, using a conductivity meter (4168, Traceable^®^ Products, Webster, TX, USA). The pH of the PBS (4.5–5.2 ± 0.2), UP water (4.8 ± 0.2), and YMBW (3.4 ± 0.2).

### 2.2. Design and Construction of the Electroporator

#### 2.2.1. A High-Voltage Generator Was Used for Applying an Electric Field to the *S. cerevisiae* Suspension

The generated voltage was adjusted to 900 V (U). The voltage pulses were controlled by a signal generator (Stanford Research System DS45, 30 MHz (Sunnyvale, CA, USA). The voltage between the electrodes of the chamber and the current was measured using an oscilloscope (Tektronix, TDS 380, 400 MHz, 2 GS/s). A voltage probe that decreased the voltage U by a factor of 1000 was used for measuring the chamber voltage U_CH_, while the current I_CH_ was measured using a Rogowski coil (an electrical device for measuring current). The current density was calculated in accordance with cross-section S (J_CH_ = I_CH_/S). The schematic electronic circuit is presented in Figure 1.

#### 2.2.2. Construction of the Electroporator Chamber

The electroporate chamber was made from two stainless-steel plates, each with a thickness of 1.5 mm (width 32.7 mm × height 33.94 mm). The lower part of each electrode included a space for attaching a crocodile hook (width 47.34 mm × height 11.6 mm). Behind the electrodes were located two copper plates, each with a thickness of 5 mm and area of 1614 mm^2^ (width 40.1 mm × height 40.26 mm). The chamber size was 13.1 mm wide and 24.26 mm high. The electrodes were tightly pressed to a Teflon frame using special clamps. The chamber gap (3.1 mm) could be filled with 350 µL up to a height of 8.7 mm (current cross-section S = 1.13 cm^2^). See Figure 2 and Appendix A.

### 2.3. Characterization of One Cycle of PEF Treatment

The electric field was applied according to a previous protocol by Emanuel et al. (2019) [15], with some changes. The yeast suspensions were exposed to an electric field of 2.9 kV cm^−1^ with a frequency of 100 Hz (f) and a rectangular pulse shape, for a duration of 10 µs (τ). The number of pulses was 5000 (n), delivered in a continuous series of 10 trains of 500 pulses each. The duration of each train was 5 s with a 2 s interval between the trains. The chamber voltage U_CH_ polarity was switched for each train, so that the next train had the opposite polarity [15]. It is important to note that the pH of the solutions was examined before and after PEF treatment, and it was found that the pHs were not chanced. Thus, electrolysis did not take place in our experiments. 

### 2.4. Electro-Pulsation Procedure for Determining the Conditions of Yeast Eradication

The yeasts were suspended in 0–0.54 mM PBS concentrations, leading to a current of 0.03 ± 0.01 −3.7 ± 0.2 A. The current densities were 0.02 ± 0.01–3.3 ± 0.1 A cm^−2^, with specific densities indicated for each experiment. The yeast suspension was exposed to PEF treatment as described in paragraph 2.3. The temperature was measured with a multimeter (VICHY, VC99) equipped with a k-type chromel–alumel thermocouple. The initial yeast suspension temperature was 22 °C, and it did not exceed 35 °C by the end of the PEF treatment. After exposure to PEF treatment, the yeast suspensions were transferred to Eppendorf tubes and incubated at 37 °C for 2 h, followed by colony-forming units (CFU) analysis. When the *S. cerevisiae* in the YMBW were exposed to PEF treatment with a regime of different cycles (1–4 cycles, each as described in paragraph 2.3), the treated culture was transferred to an Eppendorf tube for a rest of 20 ± 3 min. This procedure was repeated as described for each experiment. The current measured on the chamber was 4 ± 0.2 A (I_CH_). After the last cycle, preservatives were added to the culture, followed by incubation for 24 h at 25 °C. 

### 2.5. Total Specific Energy

The total specific energy (*W_T_*) was calculated as described in the work of Raso et al. (2016) [46]. Briefly, the first stage was to calculate the specific energy input per pulse (W). The W is the integral over time of the recorded pulse shape of voltage and current that was measured on the treatment chamber during the pulse (τ).

Equation (1):(1)$$W=\frac{1}{m}\int_{0}^{\infty }U\left(t\right)\cdot I\left(t\right)dt\;,$$
where m is the sample mass, U(t) is the voltage, and I(t) is the current measured on the PEF chamber during load pulse (τ). The total specific energy (*W_T_*) for each treatment was determined by multiplying the pulses number (*n*) with specific energy per pulse (W).

Equation (2):(2)$$W_{T}=W\cdot n$$

In this research, the number of pulses was 5000, but the energy per pulse changes in correlation to the current that was generated in line with the culture’s conductivity.

### 2.6. Electro-Pulsation Procedure for Determination of Membrane Permeability and Viability as a Function of Dilution in PBS and YMB Media

The yeast suspensions (700 µL) in 0.54 mM PBS were exposed to PEF, as described in Section 2.3. The yeast suspension was then divided into three parts. One portion was examined immediately after the PEF treatment by flow cytometry (FCM) analysis and viable count assay. The second and the third portions were diluted (1:10) in 0.54 mM PBS or YMB medium and incubated for 24 h at 37 °C. These yeast suspensions were sampled for FCM and CFU mL^−1^ analyses as described by Emanuel et al. (2019) [15]. The same procedure was conducted for the control samples, but without exposing the yeast suspension to PEF treatment. 

### 2.7. Examination of Yeast Membrane Permeability Using FCM Analysis

An electro-treated yeast suspension and the nontreated control were diluted 10-fold in 0.54 mM PBS and YMB, followed by staining with fluorescent propidium iodide (PI), as described by Emanuel et al. (2019) [15]. 

### 2.8. Statistics

Data are expressed as means ± SE (standard error) of between three and five replications. The paired *t* test was used for estimation of statistical significance. The results were considered statistically significant at *p* < 0.05. 

## 3. Results and Discussion

### 3.1. The Effect of the Current Density on S. cerevisiae Eradication 

The *S. cerevisiae* (0.01 OD 600 nm) in 0–0.54 mM PBS with a conductivity range of 1–1050 µS were examined in an electric field of 2.9 kV cm^−1^. The yeast suspension conductivity influenced the generated current density, which was between 0.02 ± 0.01 and 3.3 ± 0.1 A cm^−2^. The electro-treated yeast suspensions were incubated for 2 h at 37 °C, and the yeast viability was examined by a viable count assay (Figure 3).

*S. cerevisiae* (0.01 OD) suspensions in 0–0.54 mM PBS were exposed to an electric field of 2.9 kV cm^-1^ with different current densities (0.02–3.3), as shown in Figure 3. PEF treatment (2.9 kV cm^−1^) of *S. cerevisiae* suspended in UP water did not influence the yeast concentration. However, when the yeasts were suspended in increasing PBS concentrations (linearly corresponding to higher medium conductivity, current density, and total specific energy), a decrease in the yeast CFU ml^−1^ was observed. Total yeast death (5.15 log10 reduction) was observed at a current density of 3.3 ± 0.1 A cm^−2^ and total specific energy of 614.59 kj kg^−1^.

The electroporation process is explained by an electromechanical model, where the membrane is considered an elastic dielectric capacitor. When external electric field pulses are delivered, the transmembrane potential rises, and the mechanical compression forces are increased [47]. The effect of medium conductivity is an important factor in this model. Pucihar et al. (2001) examined PEF treatments on cells with different medium conductivities of 0.001–1.6 S m^−1^ and found that the percentage of cell death increased with the rise in medium conductivity [22]. 

A study by Ou et al. (2016) [48] examined the effect of different parameters, such as electric-field strength, pulse width, and specific energy input, on membrane permeabilization and inactivation efficiency when using PEF on *S. cerevisiae* yeast. Their results showed that the specific energy input was the major parameter influencing cell survival [48]. Timmermans et al. (2019) studied the effect of electric-field strength (E) and pulse width (τ) on various microorganisms, including *S. cerevisiae*. They found that PEF treatment at E = 2.7 kV cm^−1^ and τ = 1000 μs led to a reduction of *S. cerevisiae* by 5 log10. This eradication occurred at lower temperatures, compared to equivalent thermal processes common in the food industry. The same effect was observed when the PEF conditions were E = 2.7 kV cm^−1^ and τ = 100 and 15 μs [49]. 

The above-mentioned studies are compatible with our results that demonstrated a decrease of cell concentration correlated to increased medium conductivity. Notably, the PEF conditions were not accompanied by high temperature, which may by itself lead to yeast death. 

Many studies have shown the influence of PEF parameters on cell survival, such as conductivity, temperature rise, and total specific energy. However, relatively few studies have shown the influence of the current density on cell death. We assume that current density can also damage cells by means of charged particles that move quickly and collide with cell membranes. Bockmann and Grubmuller (2004) reported that different types of ions have different damaging effects on the cell membrane. The distinction between divalent or monovalent cations (often anionic) can cause a different number of bonds with the phospholipid headgroups [50]. These bonds can change the moment dipole and net tilt of headgroups. Moreover, the ion’s size and charge also have an important influence on the direction and magnitude of these shifts [51]. Muraji et al. studied the influence of different extracellular ions on *S. cerevisiae* membrane permeability. Equally conductive solutions with different ion compositions (MgCl_2_, NaCl, KCl, MgSO_4_ or CaCl_2_) were used for suspensions of *S. cerevisiae*. It was shown that when the cells were suspended in a solution containing NaCl, a larger permeabilization was observed in comparison to the other ions [52].

### 3.2. Viability of PEF-Treated S. cerevisiae as a Function of Suspension in YMB Medium vs. PBS

*S. cerevisiae* were treated by PEF (2.9 kV cm^−1^, current density of 3.3 ± 0.2 A cm^−2^, 100 Hz, 5000 pulses of 10 µs each), and their concentration was examined during a time period of 24 h. 

The PEF-treated yeast (0.02 OD 600 nm) was divided into three parts. A suspension of 100 µL was immediately examined for CFU concentration (time ‘0’). The second suspension of 100 µL was diluted in 900 µL of PBS (0.54 mM), defined as PEF-treated yeast in PBS, and the third suspension of 100 µL was diluted in 900 µL YMB and was defined as PEF-treated yeast in YMB. The untreated PEF yeast suspensions were treated the same way but without exposure to PEF, defined as nontreated yeast in YMB and nontreated yeast in PBS. All the samples were incubated at 37 °C for 24 h, and at indicated times, a viable count assay was performed (Figure 4).

The PEF-treated yeast concentration at time ‘0’ was 1.01·10^3^, while the nontreated yeast sample exhibited 2.35 × 10^5^ CFU mL^−1^ (a decrease of 2.36 × log10). The concentration of the nontreated yeast in PBS did not change during the entire experiment (24 h), and the values for CFU mL^−1^ were about 2.10 × 10^5^. A continued replication in the nontreated yeast in YMB was observed, and after 24 h it reached 4.40 × 10^7^ CFU mL^−1^. The concentration of the PEF-treated yeast in PBS after 1 and 2 h were 7.5 × 10^2^ and 8 × 10^2^ CFU mL^−1^, respectively. However, no CFUs were observed 5 h after the PEF treatment, which held true until the end of the experiment. The PEF-treated yeasts in YMB were not replicated for about 2 h, remaining at 1.16 × 10^3^ CFU mL^−1^. However, after 2 h the cells continued to replicate and reached 1.66 × 10^5^ CFU mL^−1^.

In conclusion, when the yeasts were suspended in PBS (0.54 mM) and treated by PEF, a reduction of 2.36 log10 in time ‘0’ was observed. No CFUs were observed at 5 and 24 h after the PEF treatment. However, the yeast cells suspended in YMB continued to replicate, and after 24 h they reached 1.66·10^5^ CFU mL^−1^. 

Recent studies have shown the existence of sublethally injured *S. cerevisiae* after exposure to PEF treatment [53]. This phenomenon depends on the treatment conditions (medium, electric-field strength, number of pulses, pulse shape, and duration, etc.). The effect of different media on the recovery of injured *S. cerevisiae* cells after PEF treatment was therefore examined. After PEF treatment (50 pulses at 12.0 kV cm^−1^ at pH 4.0 or 7.0), sublethal injury of yeast cells was detected. Use of Sabouraud dextrose broth led to maximum repair of the injured yeasts, compared to citrate–phosphate buffer (pH 7.0) or peptone water (pH 4.0). In citrate–phosphate buffer, no repair was observed, while a higher extent of PEF-treated *S. cerevisiae* repair and survival was observed in the acid conditions [34]. 

The recovery of sublethally injured *S. cerevisiae* has also been reported when the cells were exposed to thermal or chemical stresses [54]. Zakhem et al. studied how PEF-treated *S. cerevisiae* are affected by moderate electric fields when E < 7.5 kV cm^−1^. The observed damage in the early stages was not complete and continued to develop a long time after the PEF treatment [55]. Our results also found a developing damage that continued for several hours after the PEF treatment. When treated yeast was suspended in PBS, a decrease in the CFU ml^−1^ was shown. However, only after 5 h was total eradication observed.

### 3.3. Membrane Permeability of PEF-Treated S. cerevisiae When Suspended in YMB vs. PBS

A yeast suspension (0.02 OD 600 nm) in 0.54 mM PBS was divided into three parts, as described in Section 3.2. The yeast suspensions were incubated at 37 °C for 24 h, and at indicated times (0, 1.5, 3, and 24 h) 1.8 µL of propidium iodide (PI) was added to a sample of 120 µL for 5 min incubation, followed by analyzing the membrane permeability by FCM (Figure 5).

As shown in Figure 5, the percentage of the PI-positive PEF-treated yeast (suspended in PBS) at time ‘0’ was 98 ± 0.3%. In contrast, the nontreated samples (both YMB and PBS) exhibited only a low PI-positive percentage of about 1.5 ± 0.1%, which was observed throughout the 24 h of the experiment. 

The PI-positive percentages of PEF-treated yeast in YMB after 1.5 and 3 h were 97 ± 0.4% and 94 ± 0.8%, respectively. Similar results were observed with PEF-treated yeast in PBS: 98 ± 0.3% and 98 ± 0.2%, respectively. In contrast, after 24 h, the PI-positive percentages of PEF-treated yeast in YMB were about 4 ± 1%, compared to 97 ± 0.3% of PEF-treated yeast in PBS. 

We assume that the low PI-positive percentages seen at time ‘24’ (PEF-treated yeast in YMB) are a result of a membrane repair mechanism in this rich medium. The same result was observed in yeast multiplication, as shown in Figure 4. Neither yeast multiplication nor membrane recovery occurred when the cells were suspended in PBS (Figure 4), and high membrane permeability was confirmed by the data in Figure 5. 

The mechanisms of prolonged membrane permeability, which continued long after pulse delivery, remain unclear. Several studies have explained the resealing process via a “molecular” mechanism such as lipid peroxidation [56,57]. In contrast, Stirke et al. (2019) studied the cell resealing process of *S. cerevisiae* yeast and suggested that the recovery process is “mechanical” in nature; it may be attributed to changes of osmotic pressure during electroporation and the processes taking place after PEF [58]. 

Our results showed that PEF-treated *S. cerevisiae* membrane permeability lasted for a few hours. This phenomenon was observed when the cells were suspended in PBS as well as in YMB. However, in the rich YMB medium membrane, recovery was observed after 24 h. We assume that the richness of the medium enabled a recovery process unattainable for cells suspended in PBS (which includes only salts). These conclusions are important for the implementation of PEF technology in the food industry.

### 3.4. Yeast Eradication as a Function of PEF Treatment-Cycle Mode 

As mentioned, the *S. cerevisiae* were grown in YMB with tartaric acid (pH 3.4) to the mid-log phase. This was followed by addition of ethanol to a final concentration of 10% (*v/v*) and 3.4 pH (designated as YMBW). After 2 h of growth following the addition of ethanol, the culture was diluted to a final concentration of 0.02 OD at 600 nm. Then the *S. cerevisiae* was exposed to PEF treatment: 2.9 kV cm^−1^ at a current density of 4 ± 02 A cm^−2^, frequency of 100 Hz, pulse duration of 10 µs, with 5000 pulses (the described PEF treatment condition was designated as “cycle”). The treated culture was transferred to an Eppendorf tube for a rest of 20 ± 3 min before exposure to another PEF cycle. The yeast culture was treated in one to four cycles, as indicated in the experiment.

The PEF-treated *S. cerevisiae* culture was immediately (“0 time”) after each PEF treatment examined for CFU mL^−1^ (Figure 6). 

As shown in Figure 6, one cycle of PEF treatment led to a reduction of 1.35 log10 compared to the control. However, increasing PEF cycles from one to four increased the yeast mortality by only a 0.89 log10 reduction. In other words, addition of PEF treatment cycles enhanced the yeast eradication, but the influence of the first PEF treatment cycle on *S. cerevisiae* eradication was more significant compared to the subsequent cycles.

### 3.5. S. cerevisiae Eradication as a Function of Combined PEF Treatment and Preservative 

The *S. cerevisiae* at their log phase in YMBW (0.02 OD at 600 nm) were exposed to different numbers of PEF treatment cycles (each cycle: 2.9 kV cm^−1^ at current density of 4 ± 0.2 A cm^−2^, frequency of 100 Hz, pulse duration of 10 µs with 5000 pulses, 20 ± 3 min break between each cycle; more details in Section 2.4). At the end of the PEF-treatment cycles, a solution of free SO_2_ and sorbic acid was added to achieve a final concentration of 25 ppm free SO_2_ and 25 ppm sorbic acid. The PEF-treated and non-treated *S. cerevisiae* with and without the preservatives incubated for 24 h at 25 °C. A viable count assay was performed at the end of incubation. 

A similar procedure was conducted for two additional sets, except for varying the added concentrations of free SO_2_ and sorbic acid. For one set, the final concentration of the treated cultures was 50 ppm free SO_2_ and 125 ppm sorbic acid. In the other set, no preservatives were added. 

The next steps of the procedure were the same as described at the beginning of this experiment (24 h of incubation at 25 °C, followed by CFU mL^−1^ analysis). Results are shown in Figure 7.

As shown in Figure 7, the control sample without the preservatives exhibited a CFU mL^-1^ that was two orders of magnitude higher than the control samples with either high or low concentrations of preservatives. Regarding length of PEF treatment (1–4 cycles), the CFU mL^−1^ values in the cultures that were only PEF-treated (where no preservatives were added) were about the same compared to the samples of yeast culture with preservative and that were exposed to PEF.

The eradication increased along with the increase in the number of PEF treatment cycles. The addition of the preservatives did not lead to higher reduction in the yeast concentration compared to the sample that was exposed to PEF without the preservatives.

In conclusion, the added PEF treatment cycles improved the eradication of *S. cerevisiae*, while a synergic effect was not observed when free SO_2_ and sorbic acid were added.

Due to the necessity of finding different techniques for inactivating microorganisms in the food industry as alternatives to heat treatments [40,50], Montanari et al. studied the combined effect of PEF and citral preservative on *S. cerevisiae* inactivation. Following citral insertion, the log reduction increased approximately by 1.5 fold, compared to PEF treatments without citral. Although synergism was not always observed, Somolinos et al. (2007) reported that membrane permeabilization induced by PEF can improve the penetration of preservatives. However, PEF treatment along with the addition of sorbic acid did not enhance the inactivation of *S. cerevisiae* in this manner when the concentration was less than 200 ppm [59].

## 4. Conclusions

When PEF treatment (2.9 kV cm^−1^) was used on *S. cerevisiae* suspended in increasing PBS concentrations (linearly corresponding to higher medium conductivity, current density, and total specific energy), a decrease in yeast concentration was observed. The type of medium (rich or PBS) was critical for the success of injured yeast recovery, or alternately for continued damage development until total eradication. Additional PEF treatment cycles improved the eradication of *S. cerevisiae*. However, a synergic effect was not observed when free SO_2_ and sorbic acid were added. This study shows the important and necessary knowledge about yeast viability and membrane recovery after PEF treatment, in particular for applicability in the liquid food industry.

## Figures and Tables

**Figure 1 microorganisms-08-01684-f001:**
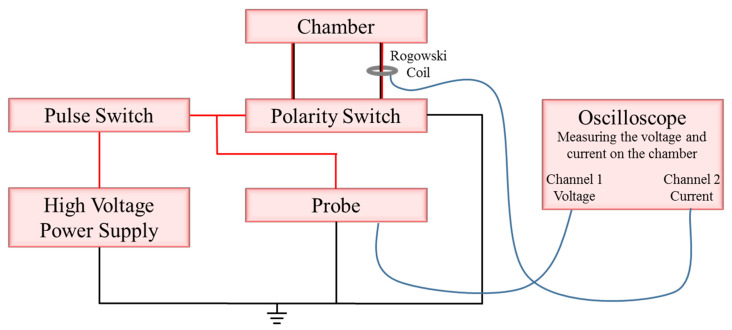
Schematic drawing of the high-voltage generator and the electronic circuit.

**Figure 2 microorganisms-08-01684-f002:**
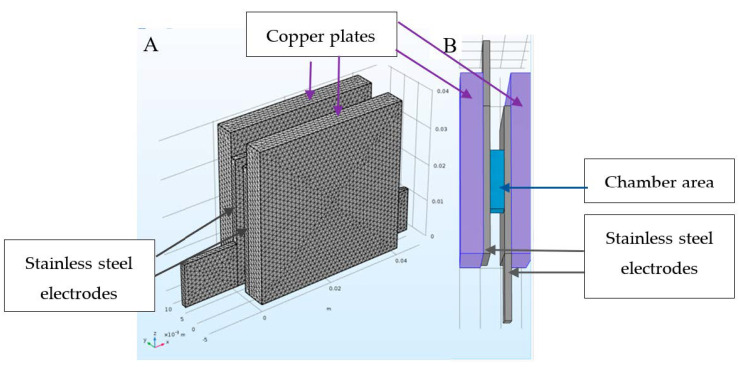
Schematic drawing of the electroporator chamber: **A**: side view, **B**: bottom view.

**Figure 3 microorganisms-08-01684-f003:**
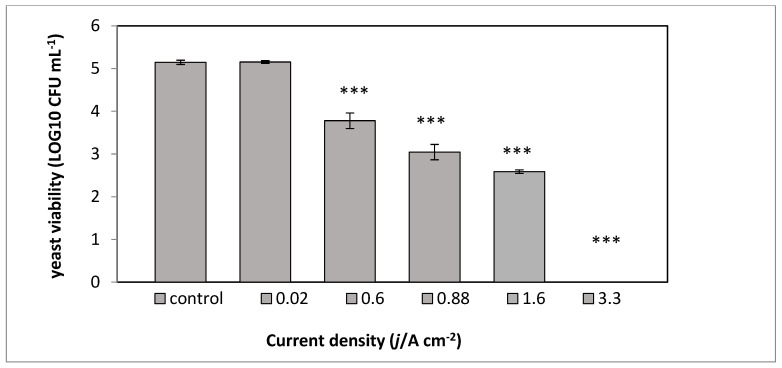
The effect of the current density on *S. cerevisiae* viability. The control consisted of yeast suspension in UP water without PEF treatment; the second column (0.02 A cm^-2^) was a yeast suspension in UP water with exposure to PEF treatment; the remaining columns (0.6–3.3 A cm^−2^) were yeast suspensions in PBS solutions with different conductivities, described in Table 1. PEF conditions: electric field intensity 2.9 kV cm^−1^, frequency 100 Hz, 5000 pulses in trains mode of 500 each, pulse duration 10 µs. *p* values (*t* test): *** *p* < 0.001.

**Figure 4 microorganisms-08-01684-f004:**
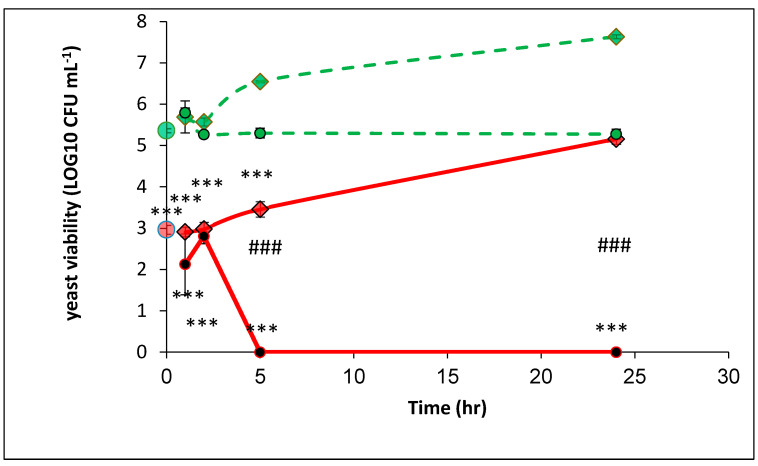
Yeast concentration of PEF-treated and nontreated. PEF-treated yeast in time ‘0’ (
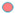
); nontreated yeast in time ‘0’ (
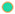
); PEF-treated yeast in YMB (1–24 h) (
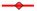
); PEF-treated yeast in PBS (1–24 h) (
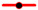
); nontreated yeast in YMB (1–24 h) (
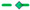
); nontreated yeast in PBS (1–24 h) (
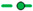
). *P* value (t test): significance of the CFU count in each examined time related to its control *** *p* < 0.001; significance of the CFU of the treated yeast in PBS related to YMB, in each examined time ### *p* < 0.001.

**Figure 5 microorganisms-08-01684-f005:**
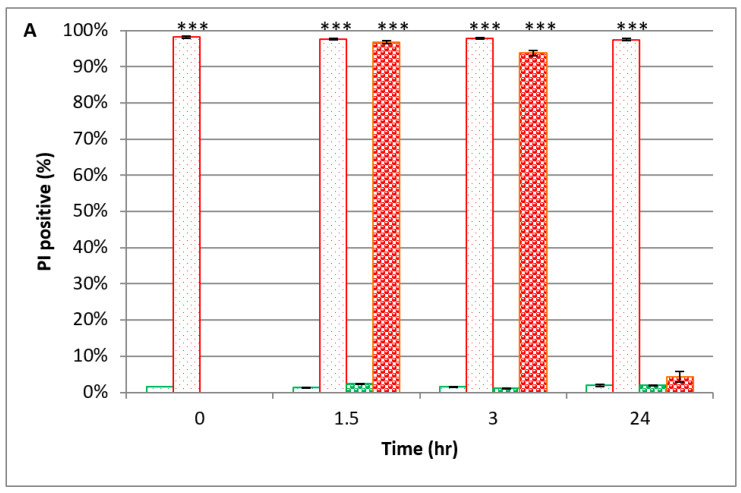
Examination of *S. cerevisiae* membrane permeability. PEF-treated yeasts, which were suspended in YMB (1.5–24 h) (
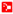
); PEF-treated yeasts, which were suspended in PBS (0–24 h) (
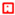
); nontreated yeasts, which were suspended in YMB (1.5–24 h) (
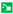
); nontreated yeasts, which were suspended in PBS (0–24 h) (
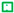
). *P* value (t test): *** *p* < 0.001.

**Figure 6 microorganisms-08-01684-f006:**
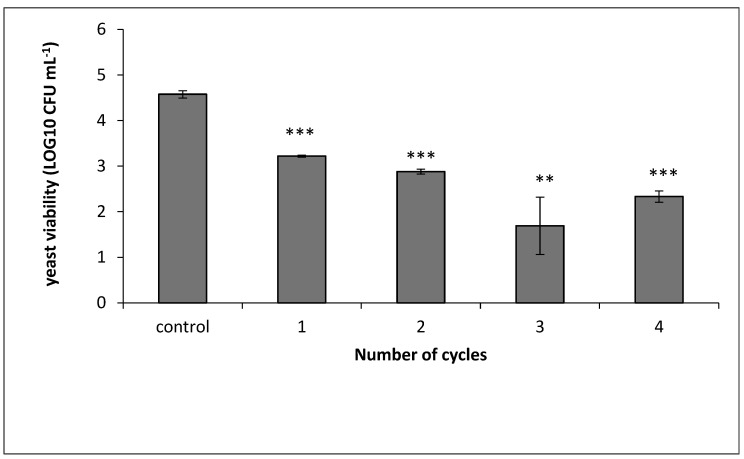
Viability of *S. cerevisiae* in YMBW as a function of PEF treatment cycles. The control column represents the CFU mL^−1^ of yeast cells without PEF treatment. Columns 1–4 are the number of treatment cycles applied to the yeast culture. The conditions of each PEF cycle: electric-field intensity 2.9 kV cm^−1^, frequency 100 Hz, pulse duration 10 µs, conductivity 1200 µS. *p* value (*t* test): ** *p* < 0.01; *** *p* < 0.001.

**Figure 7 microorganisms-08-01684-f007:**
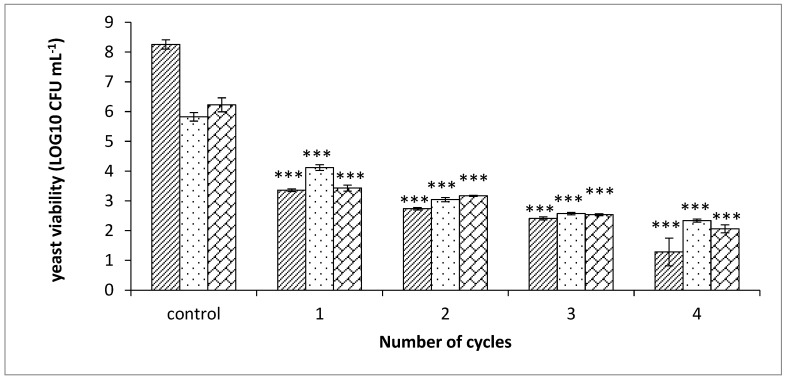
Viability of *S. cerevisiae* in YMBW after 24 h of incubation at 25 °C, as a function of PEF treatment cycle numbers and added preservatives. The control columns represent the CFU mL^−1^ of yeast without PEF treatment. Column groups 1–4 represent the number of cycles that were applied to the yeast culture. No added preservatives (
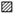
); added 25 ppm free SO_2_ and 25 ppm sorbic acid (
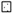
); added 50 ppm free SO_2_ and 125 ppm sorbic acid (
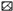
). The conditions of each PEF cycle: electric field intensity 2.9 kV cm^−1^, frequency 100 Hz, pulse duration 10 µs, conductivity 1200 µS cm^−1^. *p* value (*t* test): *** *p* < 0.001.

**Table 1 microorganisms-08-01684-t001:** Measured and calculated values as a function of PEF treatment in different media with *S. cerevisiae.*

*j* (A cm^−2^)	λ_0_ (µS cm^−1^)	*W_T_* (kJ kg^−1^)	∆T_av_ (°C)
control	1	--------	--------
0.02	1	22.04	0 ± 0.1
0.6	155	135.06	ND
0.88	330	136.38	ND
1.6	540	275.74	6 ± 0.1
3.3	1030	614.59	9 ± 0.1

λ_0_—medium conductivity, *j*—current density, *W_T_*—total specific energy, ∆T_av_—average temperature rise. ND—no data.

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
