# Peer review of "Eradication of Saccharomyces cerevisiae by Pulsed Electric Field Treatments"

_microorganisms, 2020, doi:10.3390/microorganisms8111684_

Round 1

Reviewer 1 Report

In this work the authors are studying the impact of Pulsed electric fields on the yeast S. cerevisiae.

They designed a valuable electroporator chamber depicted in figure 2. I think there is a problem with the arrows in this figures, some of them pointing "nothing" or obvioulsy not what they should.

the authors analyzed the effect of the current density on the cells viability, but also the effect of the cycles number, and of preservative products (ie SO2 and ascorbic acid). I think these 3 histograms could be lerge into only one. In these 3 histograms, I have the same concern. The statistical comparison are only performed agaisnt the control (as far as I understand). So it is difficult to conclude on the effetc of increasing the current density for exemple. Current conditions should be compared with each others. same for the number of cycle and the presevative. The corresponding text need to be modify in regard to the results of thes comparison.  In the current form, for exemple, i do't think that you can wirte, line 233-234 that "the decrease S. cerevisiae concentration was correlated to the current". You need to staistically prove it.
More over figure 3 presents CFU results, so it is not the absolute concentration, but the concentration of cells that accpeted to grow in the time and environment provided.

Globally the experiments are well designed and carrefully conducted. The paper is of undeniable merit and should be published after some revisions as detailed above.

Author Response

We would like to thank the reviewer for the helpful comments which helped us to improve the manuscript

In this work the authors are studying the impact of Pulsed electric fields on the yeast S. cerevisiae.

  • They designed a valuable electroporator chamber depicted in figure 2. I think there is a problem with the arrows in this figures, some of them pointing "nothing" or obvioulsy not what they should.

Author reply: The arrows were corrected

  • the authors analyzed the effect of the current density on the cells viability, but also the effect of the cycles number, and of preservative products (ie SO2 and ascorbic acid). I think these 3 histograms could be lerge into only one.

Author reply: The three figures are different.

Fig 3. The CFU/ml were examined as a function of the current density which was correlated to the solution (PBS) conductivity.

Fig 6. The CFU/ml were examined as a function of the number of treatment cycles while the solution conductivity was the same in all the samples. In addition, the medium was not PBS but a rich medium that mimic wine. The CFU/ml were examined immediately after each treatment.

Fig 7. The CFU/ml were examined as a function of the number of treatment cycles and in the presence of preservatives. In addition, the CFU/ml were examined after incubation of 24 hours 

  • In these 3 histograms, I have the same concern. The statistical comparison are only performed agaisnt the control (as far as I understand). So it is difficult to conclude on the effetc of increasing the current density for exemple. Current conditions should be compared with each others. same for the number of cycle and the presevative. The corresponding text need to be modify in regard to the results of thes comparison.  In the current form, for exemple, i do't think that you can wirte, line 233-234 that "the decrease S. cerevisiae concentration was correlated to the current". You need to staistically prove it.

Author reply: We have done t-test and compared the significance between the control and each of the tested experiment group.

In Fig 3 it can obviously see that the decrease of the S. cerevisiae concentration was correlated to the current density

In fig 6 all the treatment cycles led to decrease in CFU compare to the control. We wrote “As shown in Figure 6, one cycle of PEF treatment led to a reduction of 1.35 log10 compared to the control. However, increasing PEF cycles from one to four increased the yeast mortality by only a 0.89 log10 reduction. In other words, addition of PEF treatment cycles enhanced the yeast eradication, but the influence of the first PEF treatment cycle on S. cerevisiae eradication was more significant compared to the subsequent cycles”.

In fig 7 we wrote“ the control sample without the preservatives exhibited a CFU mL-1 that was 2 orders of magnitude higher than the control samples with either high or low concentrations of preservatives”. Regarding length of PEF treatment (1-4 cycles), the CFU mL-1 values in the cultures with only PEF-treated (where no preservatives were added) were about the same compared to the samples of yeast culture with preservative and that were exposed to PEF.

The eradication increased along with the increase in the number of PEF treatment cycles. The addition of the preservatives did not lead to higher reduction in the yeast concentration compared to the sample that was exposed to PEF without the preservatives.

In conclusion, the added PEF treatment cycles improved the eradication of S. cerevisiae, while a synergic effect was not observed when free SO2 and sorbic acid were added.

 We do not think that in those cases the statistics between the groups are important.

 Let us know if we need to do the statistics between the groups. Please suggest which kind of statistic test to do. It will take us a few days to do it because we will send it to be done by an expert in  statistic

  • More over figure 3 presents CFU results, so it is not the absolute concentration, but the concentration of cells that accpeted to grow in the time and environment provided.

    Author reply: concentration was corrected to CFU/ ml

Globally the experiments are well designed and carrefully conducted. The paper is of undeniable merit and should be published after some revisions as detailed above.

Reviewer 2 Report

 The subject is very old. Unfoturnatelly we cannot see any novelty.

Author Response

Reviewer 2

  • The subject is very old. Unfoturnatelly we cannot see any novelty.

Author reply: we think that this subject is very important and may have a positive reflection on the food industry

Reviewer 3 Report

The manuscript is well written and sounds very attractive for the readers of this journal. The most critical comment is concerning to the design of the electroporator. I suggest to add a photos of the instrument and the shape (voltage and current) of pulse on the cuvette filled with YMB and PBS. Moreover, there is some errors into the references and check the legends of Fig.  5 and Fig. 7., please.

Author Response

We would like to thank the reviewer for the helpful comments which helped us to improve the manuscript

      The manuscript is well written and sounds very attractive for the readers of this journal.

  • The most critical comment is concerning to the design of the electroporator. I suggest to add a photos of the instrument and the shape (voltage and current) of pulse on the cuvette filled with YMB and PBS.

Author reply: We add to supplementary materials photos of the electroporator and the shape (voltage and current) of pulse (supplementary material S1)

  • Moreover, there is some errors into the references

Author reply: the references were corrected

  • check the legends of Fig.  5 and Fig. 7., please.

      Author reply: corrected

Reviewer 4 Report

This paper describes an inactivation of bacteria (budding yeast) by pulsed electric field treatment under some different media conditions. There are many reports about inactivation of bacteria by pulsed electric field treatment. However, the submitted paper clearly shows the effect of the current density on inactivation efficiency under some different media conditions.  The results include useful information to the researchers in same research fields. The study is based on original work. Followings are comments for improvement of paper quality;

  1. Conversion errors: There are some spots in the text that seem to be conversion errors. In the figure 2, the points where the arrow points are slightly out of alignment. The sentence “Error! Reference source not found” is appeared at lines; 290, 332 and 338. In the figure captions of Figs. 5 and 7, the symbols in the legend are displaced from the sentence.

Followings are minor comments for improvement of paper quality;

  1. Line 51: The brackets “(12, 19)” should be changed as “[12, 19]”.
  2. Line 230, Table 1: The temperature rise for different current densities at same electric field is an important information. Please supply the description of temperature rise for each current density.
  3. Line 303, 304, 306, 307, 311: The expression of “E+07, E+02, E+03, and E+05” is not in common usage. Please change to the common usage “107, 102, 103, and 105”.

Author Response

We would like to thank the reviewer for the helpful comments which helped us to improve the manuscript

This paper describes an inactivation of bacteria (budding yeast) by pulsed electric field treatment under some different media conditions. There are many reports about inactivation of bacteria by pulsed electric field treatment. However, the submitted paper clearly shows the effect of the current density on inactivation efficiency under some different media conditions.  The results include useful information to the researchers in same research fields. The study is based on original work. Followings are comments for improvement of paper quality;

  • Conversion errors: There are some spots in the text that seem to be conversion errors. In the figure 2, the points where the arrow points are slightly out of alignment. The sentence “Error! Reference source not found” is appeared at lines; 290, 332 and 338. In the figure captions of Figs. 5 and 7, the symbols in the legend are displaced from the sentence.

Author reply: corrected

Followings are minor comments for improvement of paper quality;

  • Line 51: The brackets “(12, 19)” should be changed as “[12, 19]”.

Author reply: corrected

  • Line 230, Table 1: The temperature rise for different current densities at same electric field is an important information. Please supply the description of temperature rise for each current density.

Author reply: a new column was inserted ('average temperature rise').

  • Line 303, 304, 306, 307, 311: The expression of “E+07, E+02, E+03, and E+05” is not in common usage. Please change to the common usage “107, 102, 103, and 105”.

Author reply: The expression of “E+07, E+02, E+03, and E+05” were changed to the common usage.

Round 2

Reviewer 2 Report

  1. In the introduction:Pls specify why the inactivation of Yeast. In wine we never inctivate yeast we inactivate all the other pathogens (bacteria, fungi,etc but not yeast.) In a previous similar paper, ref. 15 you describe the inactivation of P.Putida  and you refer to other authors  for E.coli
  2. What is the improvements of your cells? ( PEF cells) since were described in Ref 15. it seems to be identical. Please avoid repetions. It can be considered as plagiarism. 
  3. Pls specify   that is no electrolysis in the high amperage of your experiments.
  4. Since ther is  a more or less similar work by  Efrat Emanuel ref 15 in other microoganisms please state the improovments.
  5. Pls reduce the size of your presentation   taking in consideration your work  ref 15
  6. Regards

Author Response

Reviewer comments and Suggestions for Authors

  1. In the introduction: Pls specify why the inactivation of Yeast. In wine we never inctivate yeast we inactivate all the other pathogens (bacteria, fungi,etc but not yeast.) Author reply: Thank you, the importance of yeast inactivation was added to the introduction.

There are wines, for example, classic sparkling-wine that, in the final production step, the yeast biomass has to be removed. Separation is mostly based on rotating and inclining the bottle until all the yeast cells settle into the bottle's neck. This method takes about 60 days (Ribéreau-Gayonetat., 2006; Berovi et al., 2014). Pretorius, 2000 described immobilization of the yeast (109 cells g-1) in beads composed of 2% calcium alginate. This method shortened the time of yeast separation and reduced energy consumption (Pretorius, 2000). Berovi et al., 2014 developed a rapid magnetic yeast separation in sparkling wine. The cells were absorbed to magnetized nanoparticles of iron oxide maghemite, including amino groups with a positive charge that promotes electrostatic absorption of the yeast cells' negatively charged surfaces. This method led to complete yeast separation within 15 min (Berovi et al., 2014).

  1. Ribéreau-Gayon, D. Dubourdieu, B. Donèche, A. Lonvaud Handbook of Enology. The Microbiology of Wine and Vinifications (2nd ed.), John Wiley & Sons, New York (2006)

I.S. Pretorius

Tailoring wine yeast for the new millennium: novel approaches to the ancient art of winemaking Yeast, 16 (2000), pp. 675-729

Marin Berovi Matjaz Berlot Slavko Kralj Darko Makovec A new method for the rapid separation of magnetized yeast in sparkling wine. Biochemical Engineering Journal.  2014, Pages 77-84

  1. What is the improvements of your cells? ( PEF cells) since were described in Ref 15. it seems to be identical. Please avoid repetions. It can be considered as plagiarism. 

Author reply: The PEF chamber is different from ref 15. In ref 15 the electrodes of the chamber were made from two stainless steel plates (3 mm thick) tightly pressed to a Teflon frame using a special clamp.

In the current manuscript, the electroporate chamber are made from two stainless-steel plates, each with a thickness of 1.5 mm (width 32.7 mm x height 33.94 mm). The lower part of each electrode included a space for attaching a crocodile hook (width 47.34 mm x height 11.6 mm). Behind the electrodes were located two copper plates, each with a thickness of 5 mm and area of 1614 mm2 (width 40.1 mm x height 40.26 mm).

In addition, the high-voltage generator and the electronic circuit were modified. The resistant was replaced by a Rogowski coil. Thus, the schematic drawing of the high-voltage generator and the electronic circuit were added.

This information is written in each of the manuscripts

  1. Pls specify   that is no electrolysis in the high amperage of your experiments.

Author reply: Thank you. The pH of the solutions was examined before and after PEF treatment, and it was found that the pHs were not chanced. Thus, electrolysis did not take place in our experiments.    

A clarification was added

  1. In a previous similar paper, ref. 15 you describe the inactivation ofPutida and you refer to other authors  for E.coli Since ther is  a more or less similar work by  Efrat Emanuel ref 15 in other microoganisms please state the improovments.

Author reply: The current manuscript is significantly different from ref 15. It is not only differed by the organisms (P. putida in ref 15 and S. cerevisiae yeast in the current manuscript) but also in the PEF treatment mode (in the current manuscript we added repeated treatments in a cycle mode).

In addition, in the current study the yeasts were grown in yeast malt broth with tartaric acid (pH 3.4) and ethanol to mimic wine and the effect of preservatives and PEF treatment were examined. In ref 15 we examined the effect of PEF in different current densities on P. putida viability

  1. Pls reduce the size of your presentation   taking in consideration your work  ref 15

Author reply: Paragraph 3.1 was shortened. We think that reducing overall manuscript will reduce the understanding of the results and the quality of the presentation.

  1. Regards

Round 3

Reviewer 2 Report

no comments